# Protocol for a cluster randomised placebo-controlled trial of adjunctive ivermectin mass drug administration for malaria control on the Bijagós Archipelago of Guinea-Bissau: the MATAMAL trial

Harry Hutchins [1], John Bradley,[1] Elizabeth Pretorius,[2] Eunice Teixeira da Silva,[3,4] Hristina Vasileva,[1,5] Robert T Jones [2], Mamadou Ousmane Ndiath,[6] Harouna dit Massire Soumare,[6] David Mabey,[1] Ernesto Jose Nante,[7] Cesario Martins,[3] James G Logan,[2,8] Hannah Slater,[9] Chris Drakeley,[5] Umberto D'Alessandro,[6] Amabelia Rodrigues,[3,4] Anna R Last[1]

For numbered affiliations see end of article.

**Correspondence to**
Dr Harry Hutchins;
harry.hutchins@lshtm.ac.uk

## ABSTRACT

**Introduction** As malaria declines, innovative tools are required to further reduce transmission and achieve elimination. Mass drug administration (MDA) of artemisinin-based combination therapy (ACT) is capable of reducing malaria transmission where coverage of control interventions is already high, though the impact is short-lived. Combining ACT with ivermectin, an oral endectocide shown to reduce vector survival, may increase its impact, while also treating ivermectin-sensitive co-endemic diseases and minimising the potential impact of ACT resistance in this context.

**Methods and analysis** MATAMAL is a cluster-randomised placebo-controlled trial. The trial is being conducted in 24 clusters on the Bijagós Archipelago, Guinea-Bissau, where the peak prevalence of *Plasmodium falciparum* (*Pf*) parasitaemia is approximately 15%. Clusters have been randomly allocated to receive MDA with dihydroartemisinin–piperaquine and either ivermectin or placebo. The primary objective is to determine whether the addition of ivermectin MDA is more effective than dihydroartemisinin–piperaquine MDA alone in reducing the prevalence of *P. falciparum* parasitaemia, measured during peak transmission season after 2 years of seasonal MDA. Secondary objectives include assessing prevalence after 1 year of MDA; malaria incidence monitored through active and passive surveillance; age-adjusted prevalence of serological markers indicating exposure to *P. falciparum* and anopheline mosquitoes; vector parous rates, species composition, population density and sporozoite rates; prevalence of vector pyrethroid resistance; prevalence of artemisinin resistance in *P. falciparum* using genomic markers; ivermectin's impact on co-endemic diseases; coverage estimates; and the safety of combined MDA.

**Ethics and dissemination** The trial has been approved by the London School of Hygiene and Tropical Medicine's Ethics Committee (UK) (19156) and the Comite Nacional de Eticas de Saude (Guinea-Bissau) (084/CNES/INASA/2020).

## STRENGTHS AND LIMITATIONS OF THIS STUDY

⇒ The use of a placebo control arm allows identification of ivermectin's effect on malaria and neglected tropical disease transmission, above that of dihydroartemisinin–piperaquine.
⇒ Blinding of participants, distributors, investigators and outcome-assessors greatly reduces the risk of bias.
⇒ The geographical separation of island clusters minimises effects of contamination and spillover between clusters.
⇒ Sample size of 200 over 24 clusters achieves power of 80% for the primary outcome measure.
⇒ The unique setting means results may not be directly generalisable elsewhere, although malaria endemicity and transmission dynamics appear similar to those elsewhere in the region.

Results will be disseminated in peer-reviewed publications and in discussion with the Bissau-Guinean Ministry of Public Health and participating communities.
**Trial registration number** NCT04844905.

## INTRODUCTION

Malaria in sub-Saharan Africa has declined dramatically since 2000, with much of the decrease due to vector control methods such as insecticide-treated nets and indoor residual spraying.[1] However, these methods are threatened by increasing insecticide resistance in vectors[2] and their limited efficacy against outdoor-biting or outdoor-resting mosquitoes.[3] There is a clear need for additional

vector-control methodologies, including novel tools or novel uses of existing tools.

Mass drug administration (MDA) with artemisinin-based combination therapy (ACT) could reduce transmission where coverage of vector-control interventions is high by impacting the human parasite reservoir,[4] reaching even asymptomatic cases, which help perpetuate transmission in such settings.[5] Although MDA temporarily reduces malaria prevalence, there is little evidence of prolonged effect[6] and modelling predicts that without additional vector-control measures, efficacy is short-lived.[7]

Dihydroartemisinin–piperaquine (DP) is an efficacious and safe antimalarial.[8–10] Piperaquine's long half-life makes it attractive for ACT MDA.[11–14] A cluster-randomised controlled trial in Zambia showed reduced community-level parasite prevalence after two rounds of DP MDA to households with at least one case, particularly in low-transmission areas.[15] DP MDA in The Gambia also resulted in a significant reduction in infection incidence; however, these gains were short-lived in higher transmission areas.[16] There is currently little evidence of DP resistance in Africa,[17] although surveillance data following MDA are lacking.[18]

Ivermectin (IVM) has been widely used in MDA campaigns against onchocerciasis and lymphatic filariasis (LF) in Africa for decades.[19–23] It is an effective endectocide and could be used in MDA to complement vector control strategies, particularly where existing measures have been maximised.[19 24] In West Africa, IVM MDA using 150 µg/kg decreased *Anopheles gambiae* survival and sporozoite rates (SR).[25] Modelling predicts that 3 consecutive days at this dosage would reduce infectious vector populations by 68% for 60 days.[26] Three daily doses of 300 µg/kg/day or 600 µg/kg/day were mosquitocidal at 28 days post treatment[27 28] and while there was a slight increase in minor adverse events (AE) at the higher dose, significantly higher doses have been safely used to treat head-lice and onchocerciasis.[20–23 29]

Several clinical trials have demonstrated that combined IVM/ACT MDA is safe, and that it remains an effective antimalarial treatment and endectocide.[27 28 30–32] IVM/ACT MDA has been shown to be more lethal to vectors than IVM MDA alone[33] and population-level transmission modelling predicts adjunctive IVM would boost the efficacy of DP MDA in reducing malaria prevalence in both high and low prevalence settings.[26 34] Clinical trial data are needed to confirm these findings.

The RIMDAMAL trial in Burkina Faso reported that communities receiving IVM-only MDA (150–200 µg/kg), given 5 times at 3 weekly intervals after a single dose of IVM and albendazole, saw reduced incidence of clinical malaria in young children compared with communities receiving the single dose alone.[31] However, independent statistical analysis has questioned these findings.[35] The MASSIV trial in The Gambia compared 2 years of IVM (300–400 µg/kg) and DP MDA on 3 consecutive days in 3 consecutive months, against no intervention.[32] It found significantly lower malaria prevalence in the intervention arm; however, it is impossible to separate the effect of IVM from DP due to the absence of MDA in the control arm. Further trials are, therefore, required not only to confirm optimal dosage and regimen for IVM/ACT MDA, but also its impact on malaria transmission.

The Bijagós Archipelago lies 50 km off the Atlantic coast of Guinea-Bissau. Eighteen of the 88 tropical islands are inhabited, supporting a population of approximately 25 000 fishermen, hunter-gatherers and subsistence farmers. There is a long dry season alternating with a rainy season (June–October). The mean temperature is 27.3 °C with little monthly variation.[36] Qualitative surveys and daily activity mapping have shown that population movement, commonly for farming or ceremonies, is more limited than in continental Guinea-Bissau, likely due to the islands' remoteness and lack of transport.[37]

Malaria transmission is highly seasonal, peaking at the end of the rains. Serial cross-sectional surveys, powered to estimate prevalence with ±3% precision, 80% power and 95% confidence, were conducted across the Archipelago prior to this trial and showed that, in 2018, qPCR prevalence of *Plasmodium falciparum* infection was 8.5% in January (95% CI 6.2 to 10.8, n=578), 12.3% in July (95% CI 9.4 to 15.3, n=486) and 17.5% in November (95% CI 16.0 to 19.1, n=2305) (manuscript in preparation). Infection is highest in the 5–14 years age group. Bed net coverage is estimated at 92% (95% CI 86% to 96%) with reportedly high usage (86%); however, intermittent preventative therapy in pregnancy (IPTp) coverage is lower and, in some areas, non-existent.[38] IRS is not used. One pilot round of seasonal malaria chemoprevention (SMC) was conducted on the islands during the 2020 rainy season. Indoor CDC light trapping indicates that *An. gambiae* sensu stricto is the predominant mosquito species during peak malaria transmission, and SR, determined by Circumsporozoite (CSP) ELISA, suggest it is the primary malaria vector.[39]

IVM-sensitive neglected tropical diseases (NTDs) such as LF, soil-transmitted helminths (STH) and scabies are co-endemic, the qPCR prevalence of any STH infection being 47.3%, for instance.[40] NTD studies and control measures have demonstrated that MDA is feasible, acceptable and effective in this setting.[41]

Malaria remains a significant public health problem on the Bijagós Archipelago despite high coverage with control measures. The discrete geographical nature of the islands and the associated limitation of population and vector movement further serve to highlight the Bijagós as an ideal study site.[42]

MATAMAL will fill an important knowledge gap as no trial has successfully assessed the effect on malaria of IVM in addition to DP MDA.[43] It will provide valuable data on the effects on co-endemic NTDs, with a view to future integrated strategies. It will complement previous studies in evaluating acceptability, feasibility and cost effectiveness, adding to the evidence-base examining IVM's role in malaria control. Data will be used to inform and

| MDA 1 | MDA 2 | MDA 3 | | Survey | | | | | | | MDA 4 | MDA 5 | MDA 6 | | Survey | | | | | NTD |
|---|---|---|---|---|---|---|---|---|---|---|---|---|---|---|---|---|---|---|---|---|
| Cohort | Cohort | Cohort | | Cohort | | | | | | | Cohort | Cohort | Cohort | | Cohort | | | | | |
| Ento | Ento | Ento | | Ento | | | | | | | | | Ento | | Ento | | | | | |
| Jul | Aug | Sep | Oct | Nov | Dec | Jan | Feb | Mar | Apr | May | Jun | Jul | Aug | Sep | Oct | Nov | Dec | Jan | Feb | Mar | Apr |
| | 2021 | | | | | | | | | | 2022 | | | | | | | | 2023 | | |

**Figure 1** Timeline of the MATAMAL clinical trial showing months of MDA, cohort surveys, Ento, Survey and NTD survey. Rainy season months indicated in dark colour. Ento, entomological surveys; MDA, mass drug administration; NTD, neglected tropical disease; Survey, major cross-sectional surveys.

update existing models of the impact of MDA on malaria transmission.

## METHODS AND ANALYSIS

### Study AIM

To determine whether adjunctive IVM MDA co-administered with DP MDA significantly reduces the population-based prevalence of *P. falciparum* parasitaemia during peak malaria transmission season compared with DP MDA alone.

### Study design

This is a quadruple-blind (participant, intervention provider, investigator and analyst) cluster-randomised placebo-controlled trial.

Twenty-four clusters have been assigned in a 1:1 ratio to one of two trial arms using restricted randomisation[44]:
1. Intervention (DP and IVM MDA).
2. Control (DP and IVM-placebo MDA).

Restriction variables included population, baseline *P. falciparum* prevalence (qPCR and RDT), vector density, SMC coverage and presence of a health centre. Of 100 000 randomisations, the final randomisation was selected from the approximately 10% satisfying the criteria.

This protocol was developed using SPIRIT reporting guidelines.[45 46] A timeline is shown in figure 1.

### Population and setting

The trial is being conducted on the Bijagós Archipelago, Guinea-Bissau (figure 2). Government projections using the last formal census (2009) estimate a population of 25 589.[47] Individual islands will constitute 15 clusters. Three larger islands will each be subdivided into 3 clusters providing 24 clusters in total (figure 3). There will be buffer zones of at least 2.2 km between settlements in different clusters using a modified fried-egg principle.[44] Only one populated island, Soga, will be excluded as its very high baseline malaria prevalence is an outlier.

All residents of the islands, defined as anyone sleeping on the island for the majority of a given month, will be invited to participate unless they meet any of the following exclusion criteria:
1. Severe illness.
2. Age under 6 months (DP).
3. Height under 90 cm or weight under 15 kg (IVM/placebo).

4. Pregnancy (any trimester) or breast feeding (IVM/placebo). Pregnancy (first trimester) (DP).
5. Known hypersensitivity to either medication.
6. Concomitant use of drugs affecting cardiac function or the corrected QT interval (DP).
7. Travel to a country endemic for *Loa loa* (IVM/placebo).

Residents excluded from MDA will still be eligible for participation in surveys.

An extensive sensitisation campaign will take place ahead of the intervention, led by community health workers (Agentes de Saude Communitaria, ASCs). Informed written consent will be obtained after providing written/spoken information (according to literacy) in participants' own language. An independent witness will sign consent forms of illiterate participants. Parents/guardians will be asked to consent on behalf of all children under 18 years. Children aged 12–17 years will be invited to sign informed assent forms. Forms and participant information sheets for MDA are included as online supplemental materials 1–4. We anticipate very low rates of refusal based on experience with previous interventions, including MDA (for trachoma), in these communities and the perceived importance of malaria.[38]

### Intervention and control

The intervention will be entire community MDA using:
1. DP (Alfasigma, Italy).
2. IVM (Laboratorio Elea Phoenix, Argentina).
3. IVM placebo (Laboratorio Elea Phoenix, Argentina).

Standard National Malaria Control Programme interventions (triennial distribution of bed nets, IPTp, case detection and treatment with artemether–lumefantrine) will continue in both arms. The intervention arm will receive DP and IVM; the control arm will receive DP and placebo (figure 4). A full course of MDA involves three sequential daily doses of both medications, given monthly in July, August and September of 2021 (year 1) and 2022 (year 2). Monthly MDA will begin in all clusters simultaneously, with at least 28 days between each round. Distribution will be conducted by ASCs in their own villages, supervised by experienced cluster-level field assistants. Participant age, sex, weight, pregnancy and eligibility will be recorded by household. Example case record forms, in English, are included in online supplemental material 5.

DP is a rapidly acting artemisinin-based schizontocidal drug used to treat uncomplicated malaria. It is safe, well tolerated, efficacious in clearing *P. falciparum* infection

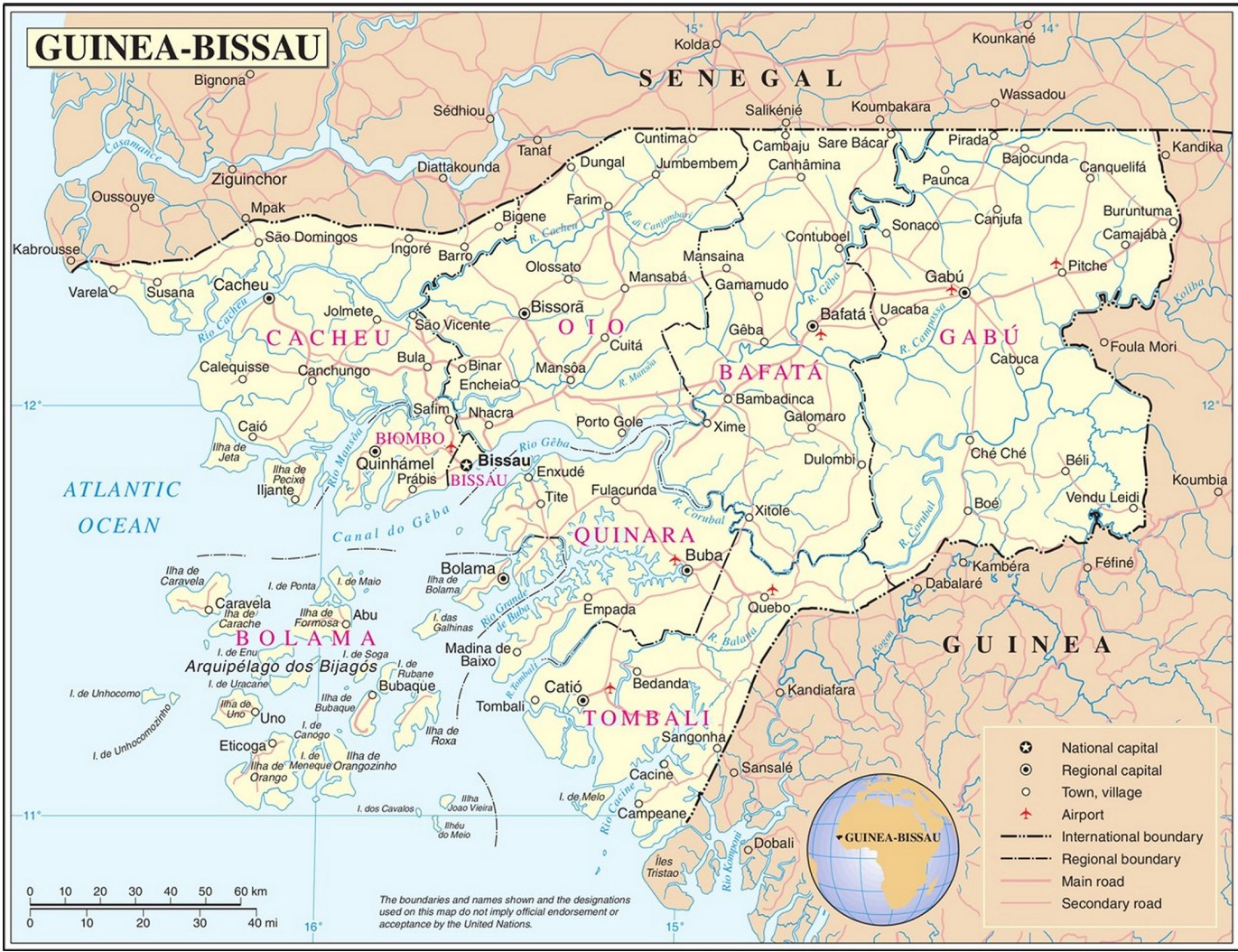

**Figure 2** Map of Guinea-Bissau showing the Bijagós Archipelago off the south-west coast.[74]]

and exhibits prophylactic activity for approximately 4 weeks.[48]

IVM is an avermectin, active against a range of human parasitic infections and infestations. It is active against *Anopheles* spp at concentrations present in human blood post ingestion.[49]

All treatment will be directly observed. Doses will be tablet formulations taken orally with water, with no food for 3 hours before or after. IVM and IVM–placebo will be visually identical 6 mg tablets dosed at 300 µg/kg to the nearest tablet. DP will be available at doses of 20/160 mg and 40/320 mg per tablet, administered according to bodyweight. DP can be crushed and mixed with water, and will be readministered in case of vomiting within 30 min (half-dose if between 30 min and 60 min).

Before delivery to the field, an independent pharmacy team will relabel all IVM and placebo bottles with identical 'IVM or placebo' labels bearing pre-printed cluster codes and recoded lot numbers to prevent unblinding. Only the pharmacist and the independent statistician who conducted the randomisation and anonymised the labels will be unblinded. All laboratory samples will bear only alphanumeric codes. Statistical analysis will be conducted by a blinded statistician. Unblinding will only occur after completion of the primary analysis.

## Outcome measures
### Primary outcome
The population-based qPCR prevalence of *P. falciparum* parasitaemia in all age groups during the peak malaria transmission season after 2 years of intervention.

This was selected as the most appropriate method for quantifying malaria transmission due to its sensitivity, especially in submicroscopic infections, the reproducibility of methods and results for this well-established assay and comparability to other trials. qPCR prevalence can also be reliably obtained from dried blood spots (DBS), a cheap, simple and robust tool with good participant acceptability.

### Secondary outcomes
1. Population-based prevalence of *P. falciparum* parasitaemia in all ages, detected by qPCR, during the peak transmission season after the first year of MDA.

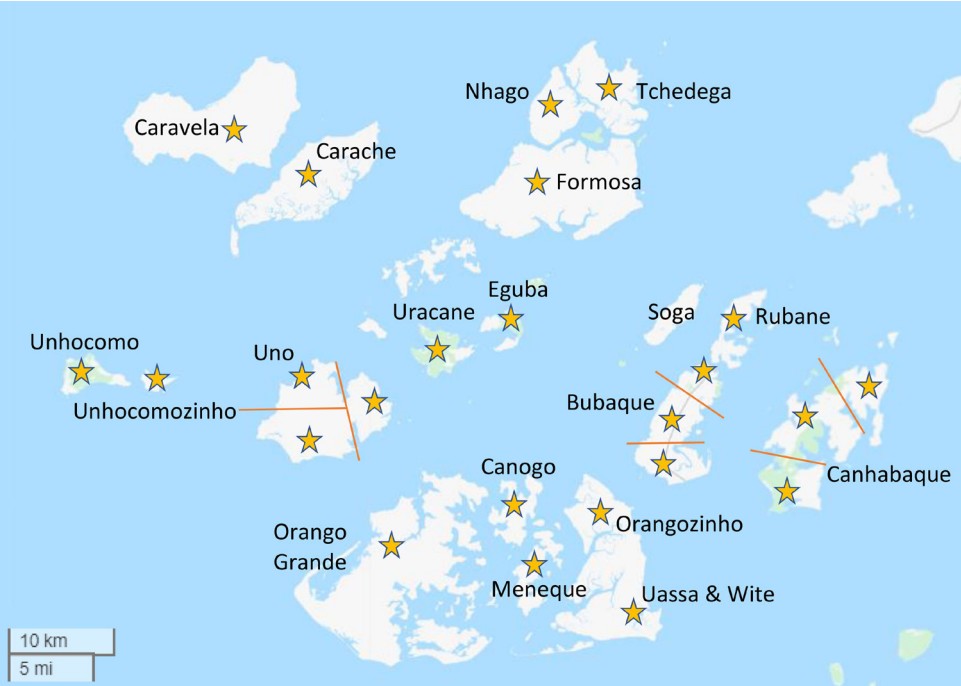

**Figure 3** Map of the Bijagós Archipelago. MATAMAL clusters marked by stars and separated by lines where sharing a landmass.

2. Incidence of clinical malaria confirmed by *Plasmodium spp.* lactate dehydrogenase/histidine-rich protein 2 (pLDH/HRP2) rapid diagnostic test (RDT), determined through passive surveillance of all malaria cases presenting to health facilities throughout the trial.

3. Incidence of clinical malaria identified by RDT (CareStart Malaria PAN pLDH) during active surveillance of a cohort of children aged 5–14 years, during the intervention and peak transmission season.

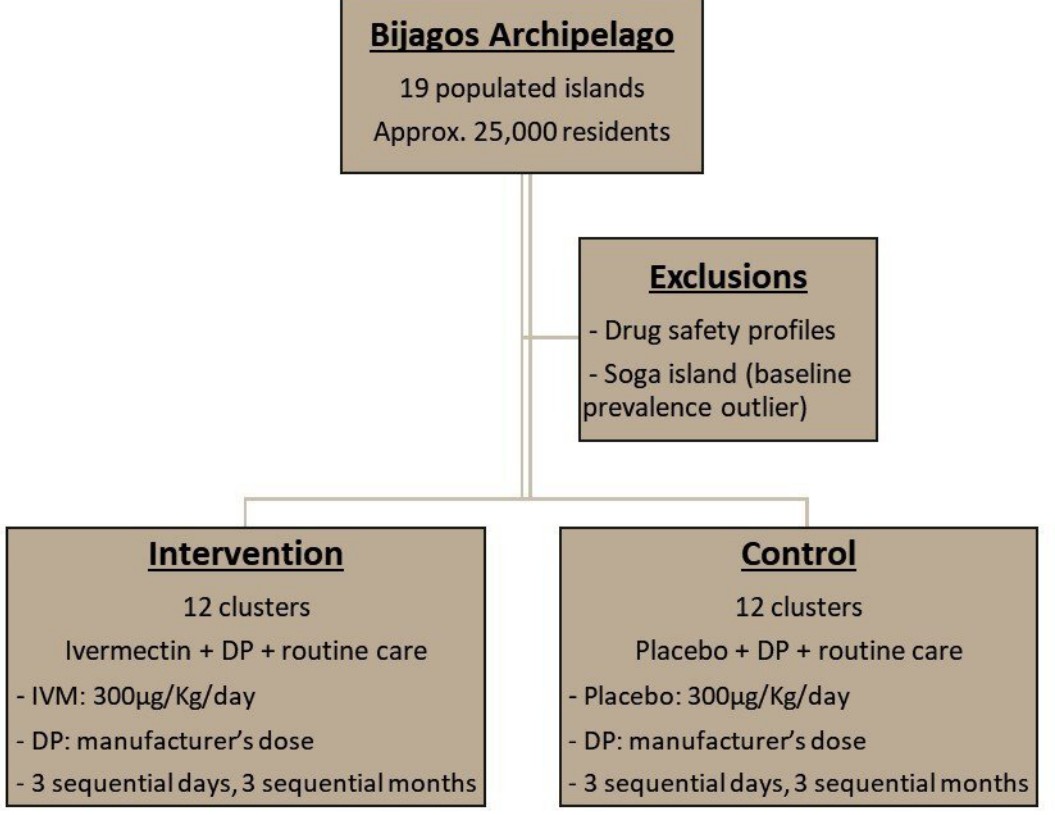

**Figure 4** Design elements of the MATAMAL clinical trial. DP, dihydroartemisinin–piperaquine; IVM, ivermectin.

4. Incidence of malaria infection identified by qPCR and serological analysis during the same period in this cohort of children.
5. Age-adjusted prevalence of serological markers indicating recent exposure to *P. falciparum.*
6. Prevalence of serological markers of recent *Anopheles* exposure.
7. Parity, as a measure of *An. gambiae* sensu latu survival, measured in mosquitoes caught using indoor CDC light traps 7–14 days after the final MDA round in year 1 and year 2.
8. Mosquito species composition, population density and SR in mosquitoes caught using indoor CDC light traps.
9. Prevalence of resistance to pyrethroids in anopheline mosquitoes using bioassay methodologies.
10. Prevalence of resistance to artemisinin and partner drugs in humans using molecular markers of resistance.
11. Safety of intervention through monitoring of AE.
12. Impact on IVM-susceptible NTDs (scabies, strongyloidiasis, other STHs and LF), headlice and bedbug infestation using clinical and serological parameters.
13. Cluster-level intervention coverage estimates.
14. MDA acceptability, feasibility and access.
15. Cost effectiveness of adjunctive IVM in this setting.

## Assessment of outcomes
### Primary
A cross-sectional survey will be conducted across all clusters beginning 4 weeks after the completion of the second year of MDA, during peak transmission. Two hundred participants will be selected by a two-step randomisation (household and individual) from within a 'yolk' of villages in each cluster, purposively defined to capture sufficiently populated villages far from other clusters but logistically feasible to reach. Socio-demographic and GPS data will be collected alongside DBS for varATS qPCR analysis, a technique capable of detecting 0.03–0.15 parasites/μL blood, 10 times the sensitivity of standard 18S rRNA qPCR.[50] Standard operating procedures are included as online supplemental materials 6–8. Participants will also be asked if they were resident in other clusters during MDA.

In November 2019, mean malaria 18S qPCR prevalence across the islands was 14.8% (95% CI 14.5% to 14.9%) (Last *et al*, mapping NTDs and Malaria on the Bijagós Archipelago of Guinea-Bissau, unpublished). These results informed a conservative estimate of the coefficient of variation of 0.46. Two hundred participants/cluster in 12 clusters/arm provides >80% power to detect a difference between arms if the primary outcome prevalence is 10% in the control arm and 5% in the intervention arm.

A mathematical model was generated to inform sample size calculations and cluster numbers. It was parameterised to simulate prevalence, seasonality, vector control and coverage. Assumptions included coverage (70% of eligible persons), DP efficacy (75% of recipients clearing

*P. falciparum* parasites), baseline peak qPCR prevalence (21.2%), dominant vector species (*An. gambiae*), fixed time lag to onward infectiousness (12.5 days from infection to gametocyte presence) and ideal conditions such as no population movement and no contamination of participants or vectors across trial arms. A validated pharmacokinetic model was used for estimating participant IVM levels over time, and mosquito mortality was assumed to increase relative to this when feeding on a given day after receiving IVM. This model predicts control arm qPCR prevalence of 9.2% after 1 year and 6.5% at 3 years, compared with 3.9% and 0.8% in the intervention arm, an effect size of 87.8% over control (figure 5).

### Secondary
1. A survey, identical to the primary outcome survey, will be conducted after 1 year of MDA.
2. On the last day of every month, data on RDT-confirmed incident malaria cases will be recorded from each of the 10 health centres on included islands: age, sex, cluster, fever, RDT performed, RDT result and treatment provided.
3. Cohorts of 50 children aged 5–14 years will be followed in 18 clusters. Households will be randomly selected from household-head lists of yolk villages. One child will be randomly selected to participate from each household, more if there are fewer than 50 eligible households. A new cohort will be selected in year 2. Participants will be visited 7–14 days after each round of MDA, as well as during the peak transmission surveys for a total of 8 timepoints. Age, sex, village, GPS and temperature will be recorded. Febrile children will be offered an RDT and treated if positive. All children will be asked about intercurrent clinical malaria episodes, trial adherence and bed net usage. A DBS will be taken for molecular and serological analysis.
4. Assessed as for outcomes 3 and 5.
5. Serological analysis will be performed on DBS taken during the cross-sectional and cohort surveys described above using a multiplex bead assay on the Luminex MAGPIX platform. The included antigens have antibodies, which are associated with recent or long-term exposure to, and/or protection from, *P. falciparum*: MSP1.19, AMA1, GLURP.R2, EBA175. RIII.V, EBA181.RIII.V, Etramp5.Ag1, Etramp4.Ag2, HSP40.Ag1, CSP, SBP1, Hyp2, GEXP18, Rh2.2030, Rh4.1 and Rh5.2.[51–53]
6. Assessed as for outcome 5.
7. One yolk village will be randomly selected per cluster. Households will be randomly selected in these villages using household-head lists (10 households/cluster in year 1; 15/cluster in year 2). A CDC light trap will be placed inside each household overnight for 3 consecutive nights, 7 days after each round of MDA in year 1 (18 clusters) and 7 days after the 3rd round of MDA in year 2 (24 clusters). There will be three further nights of trapping during the peak

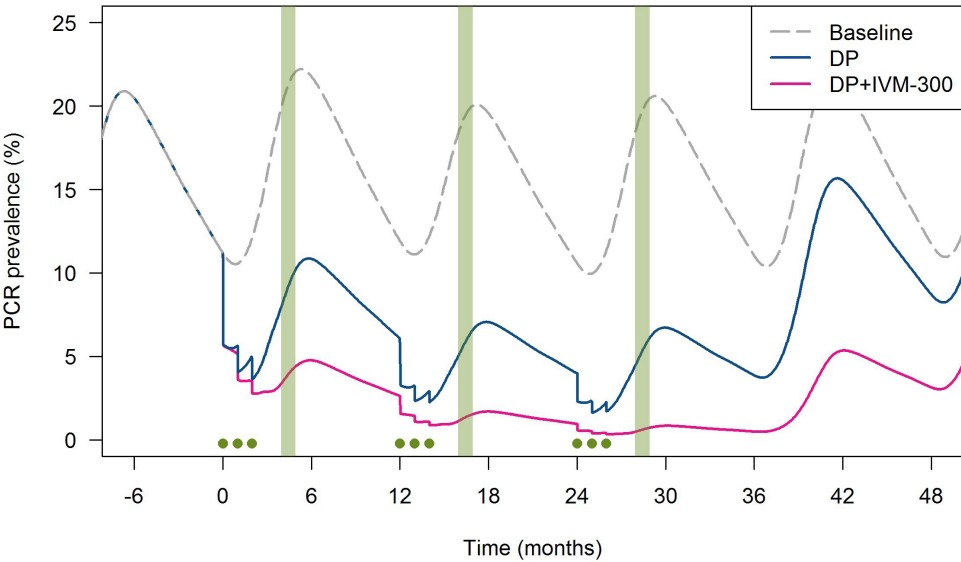

**Figure 5** Graph showing modelled qPCR prevalence of malaria over time, assuming 70% coverage and 75% DP efficacy. Green dots: MDA rounds. Green bars: survey periods. Dashed line: no intervention. Blue line: MDA with DP only. Pink line: MDA with DP and IVM at 300 µg/Kg. DP, dihydroartemisinin–piperaquine; IVM, ivermectin; MDA, mass drug administration.

transmission survey. *An. gambiae* s.l. will be identified morphologically.[54] Parity, estimating mosquito survival, will be done using the Detinova ovarian tracheation method.[55]

8. These mosquitoes will also be used to estimate cluster-level and trial arm-level species composition (proportion of species present within *An. gambiae* complex), population density (number of mosquitoes/trap/night and a proxy for human biting rate) and sporozoite detection using CSP ELISA.[56] The entomological inoculation rate (EIR), a proxy for human exposure to infectious mosquitoes and a key indicator of local transmission, will be estimated using these data.[57]

9. Anopheline mosquitoes will be reared from locally caught larvae during the year 2 rainy season. Batches of 100 adult females will be tested for resistance to alpha-cypermethrin, permethrin and deltamethrin, with and without the synergist piperonyl butoxide, using CDC bottle bioassays and WHO test tubes at once, two and five times the discriminating dose.[58 59] Two hundred mosquitoes, including all resistant individuals, will undergo PCR species identification.[60]

10. DBS samples positive for *P. falciparum* DNA will undergo further extraction, amplification by PCR[61] and sequencing using Illumina-based technology to identify genetic mutations previously associated with antimalarial drug resistance, including polymorphisms in *pfkelch*, *pfmdr1*, *pfcrt* and *pfexo* genes, before being analysed using bioinformatic tools.[62 63] Selective whole genome amplification steps will be conducted on the initial DNA extract to increase the quantity of *P. falciparum* DNA. A cohort of RDT-positive malaria patients will also be recruited from select health

centres during the rainy season to provide serial DBS throughout treatment. These will then undergo qPCR analysis for evidence of treatment success/ failure and identification of resistance markers as described above.

11. AE data will be collected actively by distributors during MDA and for 48 hours after, and passively by sensitising participants and health centre nurses to report AEs. ASCs will monitor births, deaths and miscarriages. Supervisors will send daily reports to the trial manager, who will generate annual reports for the Data Safety Monitoring Board (DSMB). Cross-sectional and cohort surveys include post hoc questions on AEs. All doses will be directly observed. *Loa loa* is not endemic to this region, greatly reducing the risk of IVM-associated encephalitis. Appropriate health-seeking advice will provided by distributors to all participants reporting AEs.

12. A population-based cross-sectional survey will be performed in all clusters during the 2023 dry season, that is, after all MDA has been delivered: 100 households will be randomly selected from household-head lists in each cluster yolk, and 1 child aged 2–10 years invited to participate from each household. Multiple children will be invited if the sample size is not met. Skin will be systematically examined to identify scabies, and impetigo in positive cases. Headlice examination and fine-tooth combing above a white cloth will occur over 10 min. Bedframes, mattresses, nets and adjacent walls will be inspected with torchlight for signs of bedbugs, or eggs, over 10 min. A stool sample (5 g) will be collected and stored in ethanol to undergo multiplex parasite PCR (STH, Strongyloides).[64–66] DBS will be collected for serological testing for LF, strongyloidiasis and STH.[67–69]

13. MDA distributors enter administration data into forms listing every householder, including absent members, and capture data on daily refusals, absences and exclusions. These forms generate a de facto census and can be used calculate monthly cluster-level coverage, absence, refusal and exclusion statistics.
14. Qualitative and quantitative analysis of MDA acceptability, feasibility and access, including population movement, will be outlined in a separate protocol.
15. Cost-effectiveness analysis will be detailed in a separate protocol.

### Data analysis plan

Data will be entered by trained MDA delivery teams under the supervision of highly trained and experienced local supervisors. Senior trial management will regularly review data entry and offer retraining as needed. Sample ID numbers will be electronically captured or double entered. A communication network will be established with remote villages to facilitate monitoring and correction of errors found on review.

All analysis will be intention-to-treat. Analysis will be performed using STATA software (StataCorp) and R.

Cluster-level, arm-level and overall demographic characteristics will be presented using descriptive statistics, including age, sex, bed net use, household size and village/cluster populations.

### Primary outcome

Trial arms will be compared using a t-test on cluster-level malaria qPCR prevalences. A risk difference, 95% CI and p value will be presented. Analyses adjusting for age groups (<5 years, 5–14 years, 15 years and above), bed net use and the presence of a health centre will be presented for this and all secondary outcomes.

### Secondary outcomes

1. As for the primary outcome, using data from the year 1 survey.
2. Cluster-level incidence rates will be compared using a t-test. If the distribution is markedly skewed, then the natural logarithm of each cluster-level summary will be taken. If some clusters have zero events, then 0.5 events will be added to each cluster. Rate differences, 95% CI and p value will be presented.
3. As for outcome 2, using RDT-confirmed incidence rates from the two annual cohorts of children.
4. As for outcome 3, using qPCR-confirmed incidence rates. Serological analysis described in outcome 5.
5. Cluster-level serological responses to malaria antigens will be presented as both continuous median fluorescent intensity (MFI) data, a proxy for antibody titres, and binary seropositivity prevalence, with a seropositivity threshold of three standard deviations above the mean malaria-naïve MFI responses. MFI will be compared between arms using mixed effect linear regression, with a random effect for cluster. Seropositivity will be compared

in the same way as the primary outcome, including analysis by age group. Cluster-level antigen-specific seroconversion rates (SCR) will be generated using serocatalytic models relying on seroprevalence, and age-seroprevalence plots will be generated by fitting data to a reversible catalytic conversion model using maximum likelihood methods. Trial arms' SCR will be compared using a t-test.[53]

6. As for outcome 5, using the gSG6 antigen and not employing SCR.
7. Cluster-level anopheline parity will be compared between arms using a t-test. Adjustment will be made for species, MDA round, time relative to MDA date, temperature, humidity and rainfall. Parity difference, 95% CI and p value will be presented.
8. The proportion of each species identified, and mosquito densities, will be presented by cluster and arm, and compared using $\chi^2$ and t-tests. Adjustment will be made for study month and year. Differences, 95% CIs and p values will be presented. If the distribution of densities is markedly skewed, a log transformation will be applied. SR will be compared between arms using a t-test on cluster-level SR with adjustments and covariates as described for parity analysis above. Risk difference, 95% CI and p value will be presented. EIR will be compared with a t-test on cluster level summaries.
9. Species composition and level of resistance will be presented using descriptive statistics. Dose–response curves will be generated for each insecticide and probit regression analysis will be performed on these data using maximum likelihood or least squares methods.
10. The number of alleles associated with antimalarial resistance will be compared over time, before and after MDA. However, appropriate statistical tests will need to be selected depending on the allele sample size before and after MDA.
11. Descriptive statistics will be presented on the number, nature, severity and relatedness of all reported AE, with additional details for serious AE. Cluster-level and age-group data will be presented. Rates will be compared between arms.
12. As for the primary outcome and outcome 5.
13. Cluster-level, arm-level and overall coverage will be presented for each month and year of intervention, and for the trial overall. The denominator for these proportions will be the total number of people recorded on MDA administration records by distributors in each cluster, whether receiving MDA or not. Refusals, exclusions and absences will be presented similarly.
14. Qualitative and quantitative analysis of MDA acceptability, feasibility and access will be outlined in a separate protocol.
15. Cost-effectiveness analysis will be detailed in a separate protocol.

## Patient and public involvement

Qualitative studies report that this population almost unanimously consider malaria to be a significant problem in their homes and the region, that additional malaria control measures would be welcome, and that MDA is acceptable,[38 70] all of which informed MATAMAL's design.

All field assistants and the deputy trial manager are local residents and MDA will be delivered by ASCs within their own communities. Methods were finalised in discussion with these stakeholders, the Regional Health Directorate and MINSAP. There will be monthly feedback sessions with ASCs and staff during MDA and surveys. Trial outcomes, social science, public engagement and qualitative work will be presented to stakeholders and communities using plans developed with the contributors themselves.

## Trial status

All necessary approvals are in place. A baseline malaria prevalence survey was completed in November 2019. Two years of MDA has successfully been delivered across all clusters, alongside two annual cohort surveys and entomological sample collection. Two annual peak transmission surveys and a dry season NTD survey have been completed. Qualitative and social science work has taken place throughout. Laboratory testing is ongoing ahead of statistical analysis.

## ETHICS AND DISSEMINATION

MATAMAL will be reported according to CONSORT guidance.[71] It is a collaboration between LSHTM, Medical Research Council The Gambia and MINSAP. Ethical approval has been obtained from LSHTM Research Ethics Committee (UK) (19156) and CNES (Guinea-Bissau) (084/CNES/INASA/2020), with additional regulatory approvals from The Gambia. Any changes to the protocol will be agreed by investigators and submitted to these same bodies for approval. The independent DSMB will oversee trial safety through annual meetings, advising the Trial Steering Committee on ongoing trial conduct. Members of these committees are listed in online supplemental material 9. Independent audit may be carried out by LSHTM, our funders or MRC The Gambia. This work abides by the Declaration of Helsinki[72] and Good Clinical Practice.[73]

Electronic forms are encrypted on submission to a LSHTM HERA-compliant secure server. Paper forms will be kept in limited-access locked storage at the site office. Data will be de-identified, except for consent and MDA record forms. All data will be held for a minimum of 7 years, including in LSHTM electronic repositories, whence access to de-identified data can be requested. All investigators will have access to cleaned databases.

Results will be published in peer-review journals, presented at conferences and to local collaborators. Field staff, nurses and ASCs will disseminate findings to participating communities in appropriate and accessible formats. Authorship will be decided on paper-by-paper basis, to recognise significant contributions to design, conduct, analysis and reporting. Writing will not be out-sourced.

**Author affiliations**
[1]Clinical Research Department, London School of Hygiene and Tropical Medicine, London, UK
[2]Department of Disease Control, London School of Hygiene and Tropical Medicine, London, UK
[3]Projecto de Saúde Bandim, Bissau, Guinea-Bissau
[4]Ministério de Saúde Pública, Bissau, Guinea-Bissau
[5]Department of Infection Biology, London School of Hygiene and Tropical Medicine, London, UK
[6]Medical Research Council Unit The Gambia, Banjul, Gambia
[7]Programa Nacional de Luta Contra o Paludismo, Ministério de Saúde, Bissau, Guinea-Bissau
[8]Arctech Innovation, London, UK
[9]PATH, Hamilton, Ontario, Canada

**Acknowledgements** We wish to express our sincere thanks to the health workers and residents of the Bijagós Archipelago, whose patience and support are the foundation of this trial.

**Contributors** ARL, the principal investigator, conceived the initial design of the study, secured funding and contributed significantly to writing this protocol. HH, the trial manager, runs all field operations, consulted on field methods and generated this manuscript. EP, RTJ and JGL drafted the entomology methods. JB provided statistical expertise. HS modelled the trial's anticipated impact. CD and HV provided serological expertise. MON and HdMS provided laboratory supervision and support. ETdS, CM, AR and EJN, our local collaborators, provided extensive information on local context and logistics. UD and DM supervised the writing of this protocol and provided expertise on trial conduct.

**Funding** This work was supported by a Joint Global Health Trials award (NIHR, MRC, Wellcome and FCDO) (funder reference: MR/S005013/1).

**Competing interests** JGL declares he is founder/CEO of Arctech Innovation, a company which aims to design mosquito lures and malaria diagnostics.

**Patient and public involvement** Patients and/or the public were involved in the design, or conduct, or reporting, or dissemination plans of this research. Refer to the Methods section for further details.

**Patient consent for publication** Not applicable.

**Provenance and peer review** Not commissioned; externally peer reviewed.

**ORCID iDs**
Harry Hutchins http://orcid.org/0000-0002-0607-6124
Robert T Jones http://orcid.org/0000-0001-6421-0881

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
