## [Reviewer comments · BMJ Open]

ARTICLE DETAILS

TITLE (PROVISIONAL)	Protocol for a cluster randomised placebo-controlled trial of adjunctive ivermectin mass drug administration for malaria control on the Bijagós Archipelago of Guinea-Bissau: The MATAMAL trial
AUTHORS	Hutchins, Harry; Bradley, John; Pretorius, Elizabeth; Teixeira da Silva, Eunice; Vasileva, Hristina; Jones, Robert; Ndiath, Mamadou; dit Massire Soumare, Harouna; Mabey, David; Nante, Ernesto; Martins, Cesario; Logan, James; Slater, Hannah; Drakeley, Chris; D'Alessandro, Umberto; Rodrigues, Amabelia; Last, Anna R.

VERSION 1 – REVIEW

REVIEWER	O'Flaherty, Katherine Burnet Institute
REVIEW RETURNED	27-Mar-2023

GENERAL COMMENTS	Hutchins et al. provide a concise and well written protocol manuscript describing an important trial of currently available tools to reduce malaria transmission. I have one query, more out of interest. What are the implications for detecting a difference in Pf prevalence between baseline and years two (and one) if the sensitivity of the varATS qPCR analysis is 10 times greater than the 18s rRNA qPCR assay used at baseline? Although this will be consistent across IVM and placebo groups, won't the baseline Pf prevalence and the effect of both arms of the intervention at years one and two be underestimated using the less sensitive assay? Are there plans to re-analyse baseline samples with the more sensitive varATS? Otherwise, I have few comments on the manuscript other than some clarification of the SPIRIT checklist entries, and I eagerly await the trial findings! I suspect during submission some page formatting may have been altered and the pages referenced in the checklist do not always match the stated section of the manuscript. I have included specific examples below, but care should be taken to check all items of the checklist. Item 11c: the page listed refers to the reference list, please correct to include the page detailing intervention adherence monitoring. Item 11d: please check the stated pages and consider whether relevant information on pages 5 and 6 should also be included to address this item. Item 14: as is, the power estimates are not described on page 5, consider including page 7 here. Additionally, Figure 3 should include the number of clusters/islands in the exclusions box. Item 16a: this information is presented on 6 not 5 as stated.
---

	Items 29 and 31c: this information is presented on page 11 not 17 as stated. Item 32: Why is this N/A if participants provided consent?
--	--

REVIEWER	Carrasquilla, Manuela Max Planck Institute for Infection Biology, Malaria Parasite Biology
REVIEW RETURNED	14-Apr-2023

GENERAL COMMENTS	I reviewed the protocol above mentioned by Hutchins et al. I was excited to read it and I consider it to be well thought out and will provide key evidence on the use of Ivermectin in addition to mass drug administration of artemisinin combination therapies. The protocol is a two-arm cluster-randomised clinical trial of mass drug administration (MDA) in the Bijagós Archipelago of Guinea-Bissau. The authors will evaluate whether the addition of ivermectin with artemisinin combination therapies (ACTs) during peak transmission would reduce Pf prevalence after two years of MDA. The authors plan to use dihydroartemisinin-piperaquine (DP) and either placebo or IVT for the control and intervention arms, respectively. It is novel and differs from previous trials in which the control arm was without DP MDA, therefore the authors consider this approach to be novel and will allow them to disentangle the impact of IVT vs DP alone. I think overall the protocol is well thought and in a geographical setting that I believe will suit the purpose of the study. I have a few comments which perhaps the authors could clarify regarding the first part: 1) It is unclear to me how the definition of limited travel within the Bijagos Archipelago and to the mainland was determined. Durrans et al 2019 (a qualitative study to evaluate human mobility across the different islands) which the authors cite, argues there is mobility primarily due to agricultural or cultural practices at different points during the year and to islands with different population sizes. How is cluster randomization accounting for movement within and between islands? 2) Is seasonality constant throughout? Are there micro-epidemiological conditions favouring e.g agricultural practices that might lead to different transmission settings and will these be accounted for? Is there travel associated for example to work in other islands that might contribute to transmission in other, more populated islands? 3) And related to the first two points, will the genotyping approach be sufficient to detect case importation from local transmission and how do you account for this in the proposed model? Further, the authors mention P. falciparum prevalence at different points of the year. For this I would like the authors to clarify a few points: 1) It is not clear from the description how prevalence was calculated, and what proportion of the population was screened by qPCR, would it be useful to know for example if this is from a cohort-based study of different ages and was there within-
---

	household confirmation of prevalence after a positive case was detected? 2) A more general question which perhaps the authors could expand a bit more on, it seems like at the start of the dry season (November) the prevalence is 17%, do the authors know what is the prevalence at the end of the dry season? Why is peak transmission a better time for MDA rather than at the start of the season when mosquitoes arrive and prevalence is likely at the lowest? Would be interesting to include in the modeling work what would happen at different times during the season. Regarding the methods and analysis: They invited all residents of the islands included in the study and they excluded one island which had the highest prevalence. This is a reasonable approach and I just have two comments: 1) It would be helpful if the authors would provide an estimate of what proportion of the population is anticipated to take part in the study and what proportion will not sign the informed consent. Is it 70-75% as stated in Figure 4? Is there a threshold of enrollment that will change the outcome of the model? For example, what happens if 80% or 50% of the population enroll in the study? I imagine the outcome will be quite different. 2) It would also be helpful if the authors include a more detailed explanation of how the modeling was done and what parameters were included and how figure 4 was generated as it is not clear from the protocol. Regarding the primary and secondary outcomes, I have the following comments: 1) The primary outcomes are clearly presented and very informative for evaluating the trial in the context of malaria elimination in the Archipelago. Given the trial will happen in addition to current interventions (in particular antimalarial interventions such as SMC), the authors could address more in depth how this will impact the landscape of resistance. Will there be any ongoing Treatment Efficacy Studies led by the MOH alongside the trial to evaluate drug efficacy? 2) I am not sure the sensitivity reported for varATS qPCR is as the authors suggest. I believe anything below 10 parasites/uL is difficult to get reliable data from (including reliable genotyping). It would be helpful to know if the authors already have a method in which they demonstrate that they can get amplification from lower parasitaemia from standard curves. The concern is that if parasitaemias in asymptomatic carriers are very low the ability to detect parasite genotypes will be reduced. 3) As in the previous point, do the authors have evidence the genotyping approach for different target regions from mosquitoes will work using an Illumina-based approach will work? They propose sWGA but this is very sensitive to DNA integrity so I am not sure of the feasibility.
--	---

REVIEWER	Kunkel, Amber Santé publique France
REVIEW RETURNED	16-Apr-2023

GENERAL COMMENTS

This protocol describes a cluster randomized control trial that is currently underway in the Bijagos Archipelago, Guinea-Bissau to evaluate the effect of ivermectin when added to mass drug administration with dihydroartemisinin-piperazine. The primary outcome is the prevalence of malaria in each arm after two years of seasonal MDA (one month after the final round). This study could play an important role in improving our understanding of ivermectin's potential role in malaria control. There is a good evidence base suggesting a theoretical impact of ivermectin on malaria transmission, it remains unclear how well this approach will perform in the real world. Furthermore, this study's design has several strengths that will help it to address this outcome, including the cluster-randomized design, the blinding of participants and investigators, and a setting which reduces the risk of cross-over effects.

The protocol provides a thorough literature review to support the importance of this trial, details on the setting where it is being conducted, and an overview of the primary and secondary outcomes including how they will be measured and assessed. My comments primarily relate to improving the clarity of the protocol. Additional review by a statistician with experience in analysis of cluster randomized control trials could be helpful, especially given the debate surrounding the analysis of the RIMDAMAL trial.

I would like to request the authors please review the entire protocol checklist and ensure all information is correct and complete, including page numbers. In several instances, the page numbers indicated do not appear to contain the specified information. For example, item #11C indicates page 14 but p.14 contains only references, and p.17 is listed as giving information on protocol amendments (#25) and plans for trial audits (#23), but I cannot find this information.

The timeline of the study is a bit unclear. For example, dates are sometimes referred to by calendar year (ex. 2022, 2023), sometimes by reference to the year of the study (ex. first year, second year). I think better consistency here would improve readability. Furthermore, a more detailed timeline figure including calendar dates, years of the study, and timing of primary and secondary outcome measurements would be helpful. This could be either combined with or in addition to Figure 4.

More information would also be useful for several topics including:

- Motivation for the choice of primary outcome
- Randomization procedure – how restrictive were the criteria applied? (e.g. of the 100,000 randomizations initially performed, how many met the criteria?)
- Procedures used to ensure data quality and plans to address any missing data
- Informed consent process including assessment of potential benefits and harms and how these were communicated to participants
- Would you be able to include forms used for data collection and informed consent as supplementary material?
- Details on the data monitoring committee
- Are any analyses planned to assess the duration of the effect?
- What is planned for malaria control in this region when the trial is over – will any intervention be maintained? How will this be decided?

	Please ensure abbreviations are defined before first use.
--	---

VERSION 1 – AUTHOR RESPONSE

Reviewer: 1 Dr. Katherine O'Flaherty, Burnet Institute

Comments to the Author:

Hutchins et al. provide a concise and well written protocol manuscript describing an important trial of currently available tools to reduce malaria transmission. I have one query, more out of interest.

What are the implications for detecting a difference in Pf prevalence between baseline and years two (and one) if the sensitivity of the varATS qPCR analysis is 10 times greater than the 18s rRNA qPCR assay used at baseline? Although this will be consistent across IVM and placebo groups, won't the baseline Pf prevalence and the effect of both arms of the intervention at years one and two be underestimated using the less sensitive assay? Are there plans to re-analyse baseline samples with the more sensitive varATS?

We used the baseline Pf qPCR prevalence data to ensure balance between trial arms at randomisation. We also compared the baseline prevalence data to previous malaria surveys conducted at this site (using the 18s rRNA qPCR assay) to look at seasonal prevalence over time (2017-2019), again to inform balance between arms in terms of stability and seasonality of prevalence data. Although of interest to look at a comparison between baseline and years 1 and 2, it isn't specifically a trial outcome. To this end, it is less important that a different assay was used at baseline, although we agree ideally the same assay should have been used throughout.

Unfortunately, we do not have the resources to re-run these samples using the varATS qPCR assay at this time, but may do so in the future for the reasons alluded to. Importantly, our outcomes focus on comparing trial arms at the same timepoints to determine impact, rather than comparing arms to baseline, and therefore the change in methodology will not affect this.

Otherwise, I have few comments on the manuscript other than some clarification of the SPIRIT checklist entries, and I eagerly await the trial findings!

I suspect during submission some page formatting may have been altered and the pages referenced in the checklist do not always match the stated section of the manuscript. I have included specific examples below, but care should be taken to check all items of the checklist.

- Item 11c: the page listed refers to the reference list, please correct to include the page detailing intervention adherence monitoring.
- Item 11d: please check the stated pages and consider whether relevant information on pages 5 and 6 should also be included to address this item.
- Item 14: as is, the power estimates are not described on page 5, consider including page 7 here.
- Item 16a: this information is presented on 6 not 5 as stated.
- Items 29 and 31c: this information is presented on page 11 not 17 as stated.

Thank you for highlighting these errors. We apologise and have corrected all formatting errors relating to the SPIRIT checklist. The listed page numbers are now correct.

- Additionally, Figure 3 should include the number of clusters/islands in the exclusions box. Soga Island, listed in figure 3, is the only excluded island. Other islands on the map not included do not have permanent populations residing there and are therefore not eligible for inclusion.

- Item 32: Why is this N/A if participants provided consent?

Apologies for this oversight when uploading documents. Informed consent forms for mass drug administration (for children under 12, adolescents aged 12-17, and adults) have been supplied as supplementary material in this latest submission.

Reviewer: 2 Dr. Manuela Carrasquilla, Max Planck Institute for Infection Biology

Comments to the Author:

I reviewed the protocol above mentioned by Hutchins et al. I was excited to read it and I consider it to be well thought out and will provide key evidence on the use of Ivermectin in addition to mass drug administration of artemisinin combination therapies.

The protocol is a two-arm cluster-randomised clinical trial of mass drug administration (MDA) in the Bijagós Archipelago of Guinea-Bissau. The authors will evaluate whether the addition of ivermectin with artemisinin combination therapies (ACTs) during peak transmission would reduce Pf prevalence after two years of MDA.

The authors plan to use dihydroartemisinin-piperazine (DP) and either placebo or IVT for the control and intervention arms, respectively. It is novel and differs from previous trials in which the control arm was without DP MDA, therefore the authors consider this approach to be novel and will allow them to disentangle the impact of IVT vs DP alone.

I think overall the protocol is well thought and in a geographical setting that I believe will suit the purpose of the study. I have a few comments which perhaps the authors could clarify regarding the first part:

1) It is unclear to me how the definition of limited travel within the Bijagos Archipelago and to the mainland was determined. Durrans et al 2019 (a qualitative study to evaluate human mobility across the different islands) which the authors cite, argues there is mobility primarily due to agricultural or cultural practices at different points during the year and to islands with different population sizes. How is cluster randomization accounting for movement within and between islands?

We have liaised closely with participating communities throughout the trial; from its design, inception and sensitisation to the delivery of the intervention, since this is crucial to the success and evaluation of a community-based intervention. The trial social science team have conducted focus group discussions, in depth interviews and participatory mapping work to understand community mobility. This body of work is the subject of a separate (but aligned) protocol to optimise intervention delivery and fidelity. This work has directly informed the implementation of the intervention in the trial, particularly as most movement appears to be predictable. In line with the findings from Durrans et al, there is temporal population mobility, primarily for reasons of agriculture, that the trial team are aware of and mitigate against (where possible) in terms of intervention delivery and outcome assessments. Knowing that some degree of movement will always occur in these populations, we are collecting data to reflect that and to adjust for in the analysis and any subsequent modelling work arising from trial data. This is also important to capture with respect to understanding how such an intervention might perform in programmatic settings. That said, compared to continental populations, there is much less movement in these island populations, with most occurring within, rather than between, clusters. We have amended the manuscript on p7, line 317 ("Participants will also be asked if they were resident in other clusters during MDA") to reflect the fact that we can estimate population movement with reasonable accuracy and adjust for it where necessary. The assessment of social science work around Acceptability, Feasibility and Access has been updated on p9, line 415 of the manuscript: "Qualitative and quantitative analysis of MDA acceptability, feasibility and access, including population movement, will be outlined in a separate protocol."

2) Is seasonality constant throughout? Are there micro-epidemiological conditions favouring e.g. agricultural practices that might lead to different transmission settings and will these be accounted for? Is there travel associated for example to work in other islands that might contribute to transmission in other, more populated islands?

We have several years of seasonal data preceding the trial suggesting that transmission is stable and seasonal as described (manuscript in preparation). Although there is some heterogeneity in prevalence between islands, accounted for in the randomisation and power calculation by the use of a conservative coefficient of variation (0.5), the environmental and epidemiological conditions throughout the archipelago are reasonably homogeneous. Population movement is captured at various points during the trial as described in the previous point.

3) And related to the first two points, will the genotyping approach be sufficient to detect case importation from local transmission and how do you account for this in the proposed model? Pathogen genomics work aiming to generate a molecular barcode for the *Plasmodium falciparum* (Pf) parasite circulating on the Bijagos Archipelago is underway. Theoretically, if enough single nucleotide polymorphisms (SNPs) can be identified, it may be possible to identify whether parasites originated from transmission within the Archipelago or from continental Guinea Bissau or beyond. This work is ongoing and beyond the scope of the trial itself, meaning that these pathogen genomic data will not be included in the primary trial analyses. In due course, through ongoing projects these data will be available and modelled in the context of these interventions. This is an area of great interest in regions near elimination, to understand the drivers of transmission following interventions.

Further, the authors mention *P. falciparum* prevalence at different points of the year. For this I would like the authors to clarify a few points:

1) It is not clear from the description how prevalence was calculated, and what proportion of the population was screened by qPCR, would it be useful to know for example if this is from a cohort-based study of different ages and was there within-household confirmation of prevalence after a positive case was detected?

The population-based prevalence of malaria (defined as the prevalence of Pf by qPCR) in all ages was estimated during cross sectional surveys conducted during peak transmission season between 2017-2019. Sample sizes were determined to estimate prevalence with +/-3% precision, 80% power and 95% confidence. This initially called for 40 individuals of all ages in each of 50 village-level clusters across the archipelago. Survey results influenced sample size calculations for later surveys. National census data were used to estimate the population on each island and a random sample of participants was selected using population proportional to size in 2017 and 2018. In 2019, 150 participants were randomly selected from randomly selected villages on each island. Dried blood spots were taken from each participant onto filter paper, and DNA extracted for qPCR as described. Annual sample sizes were 1649 (2017), 2267 (2018) and 2270 (2019). Further surveys were conducted opportunistically in different seasons within this timeframe. We have edited the relevant paragraph in the Introduction to include a summarised version of this information (p4, line 146).

2) A more general question which perhaps the authors could expand a bit more on, it seems like at the start of the dry season (November) the prevalence is 17%, do the authors know what is the prevalence at the end of the dry season? Why is peak transmission a better time for MDA rather than at the start of the season when mosquitoes arrive, and prevalence is likely at the lowest? Would be interesting to include in the modelling work what would happen at different times during the season. The primary outcome for the trial (peak transmission prevalence) is measured in November. This would be the time where we would expect to see the greatest impact following a seasonal intervention in a seasonal malaria transmission setting. The MDA schedule is indeed timed prior to that (July, August, September) to maximise impact for reasons alluded to above. In many settings, Seasonal Malaria Chemoprophylaxis (SMC), endorsed by the WHO and conducted programmatically, follows the same principle, whereby children aged 3-59 months are given antimalarials in July/August/September/October to reduce malaria incidence and mortality thereafter. Using this (and other trial data) we may be able to model the impact of this MDA regime subsequently, but this is currently beyond the scope of the primary outcomes of the trial. These trial data will be used to inform and update models originally developed by Slater et al in 2014 and 2020.[1,2] The text in the manuscript has been amended to clarify this (p4, line 174).

Regarding the methods and analysis: They invited all residents of the islands included in the study and they excluded one island which had the highest prevalence. This is a reasonable approach and I just have two comments:

1) It would be helpful if the authors would provide an estimate of what proportion of the population is anticipated to take part in the study and what proportion will not sign the informed consent. Is it 70-

75% as stated in Figure 4? Is there a threshold of enrolment that will change the outcome of the model? For example, what happens if 80% or 50% of the population enrol in the study? I imagine the outcome will be quite different.

We anticipated that the refusal rate would be very low in this study population, based on our experience of working in these communities, including distributing MDA for trachoma in previous studies. A line has been added to this effect (p5, line 225). During the first year, the refusal rate by month was 0.19%, 1.30% and 1.93%, supporting our assumption. We conducted an extensive sensitisation programme ahead of delivering the intervention in both years in order to reduce community concern about MDA. This was delivered by trained local community health workers who then went on to deliver the MDA, providing a direct link to the communities involved. This information has been added to the text (p5, line 219). Absenteeism was expected to cause the greatest reduction in coverage, the majority of which would be due to individuals spending extended periods of time away from their cluster (i.e. months), who are therefore likely not contributing to local malaria transmission, and who could not be reached even by more intensive distribution methods. As described, we are conducting surveys on population movement to be able to describe these patterns accurately in our analysis, and to improve our estimates of coverage. The model in Figure 5 assumes coverage of 70% of eligible individuals, and we believe this reflects a realistic coverage statistic when refusals and absenteeism are accounted for (p7, line 326). This coverage was also selected conservatively to ensure robust sample size calculations for outcome surveys: using higher coverage estimates would increase the predicted effect size[1] but could result in under-powered sample sizes being selected. Similarly, its conservative assumptions mean that if drug efficacy is lower than expected, the trial remains powered to detect a difference between arms.

2) It would also be helpful if the authors include a more detailed explanation of how the modelling was done and what parameters were included and how figure 4 was generated as it is not clear from the protocol.

The pre-trial model was built upon work carried out by Slater et al in 2014[1] and updated in 2020.[2] This model was parametrised to simulate prevalence, seasonality, vector control and coverage, with the aim of informing trial protocol generation, including sample size calculations and cluster numbers. As with all models, certain assumptions about trial conditions were made, including coverage (70% of eligible persons), DP efficacy (75% of recipients would clear Pf parasites), baseline PCR prevalence (21.2% at annual peak), dominant vector species (*Anopheles gambiae*), fixed time lag to a participant's onward infectiousness (12.5 days from infection to gametocyte presence) and ideal conditions such as no population movement and no contamination of participants or vectors across trial arms. A validated pharmacokinetic model was used for estimating participant ivermectin levels over time, and mosquito mortality was assumed to increase relative to this when feeding on a given day after receiving ivermectin. Assumptions regarding the impact on sporogony and mosquito feeding of ivermectin have previously been shown to be negligible. The model predicted that DP alone would reduce peak prevalence from 21.2% to 9.2% after one year of MDA, and prevalence would be 6.5% three years after starting MDA, an effect size of 68.2% over baseline. The addition of ivermectin MDA would reduce prevalence to 6.5% and 0.8% respectively, an effect size of 96.1% over baseline, or 87.8% greater than DP MDA alone. The trial would remain sufficiently powered if the ivermectin effect size were reduced to only 40%, demonstrating how the use of conservative estimations provides robust sample sizes for this trial. This information has been added to the manuscript (p7, line 326)

Regarding the primary and secondary outcomes, I have the following comments:

1) The primary outcomes are clearly presented and very informative for evaluating the trial in the context of malaria elimination in the Archipelago. Given the trial will happen in addition to current interventions (in particular antimalarial interventions such as SMC), the authors could address more in depth how this will impact the landscape of resistance. Will there be any ongoing Treatment Efficacy Studies led by the MOH alongside the trial to evaluate drug efficacy?

Thank you for this important and relevant question. SMC is not currently given on the Bijagos Archipelago. It was given previously as a pilot in 2020, and is likely to be given again following completion of the trial, depending on trial outcomes. The updated WHO recommendations for malaria chemoprevention[3] endorse much broader use of community-based administration (including MDA) of antimalarials. Whilst treatment efficacy studies (TES) are beyond the scope of the trial, we are working closely with the National Malaria Control Programme in Guinea Bissau to conduct surveillance to this effect. There have been and will be TES monitoring, and we are collaboratively conducting a TES and Pf genomic studies following MATAMAL's distribution of dihydroartemisinin-piperaquine to understand the impact of this distribution on genotypic and phenotypic resistance. It is unclear at present what the full impact of this recommendation for the expanded use of antimalarials (particularly artemisinin-based combination therapy) will be, and it is likely to vary between settings. We do plan to continue surveillance of malaria epidemiology (including antimalarial resistance) and vector ecology in this setting following completion of MATAMAL. These are areas aligned to but separate to the main trial protocol.

2) I am not sure the sensitivity reported for varATS qPCR is as the authors suggest. I believe anything below 10 parasites/uL is difficult to get reliable data from (including reliable genotyping). It would be helpful to know if the authors already have a method in which they demonstrate that they can get amplification from lower parasitaemia from standard curves. The concern is that if parasitaemias in asymptomatic carriers are very low the ability to detect parasite genotypes will be reduced. The sensitivity quoted in our protocol comes directly from Hofmann et al's method paper, where they were able to detect infections with fewer than 1 parasite/microlitre in samples of similar volume to that which is being used in MATAMAL. Importantly, varATS is more effective at detecting submicroscopic infections than the 18S method, which missed 16% of positive samples in that study. This is important in the evaluation of a community based intervention in a setting where most infections are asymptomatic and submicroscopic. We recognise that submicroscopic infections are difficult to diagnose and that it is more difficult to use these small concentrations for genomic testing, which is currently a limitation for pathogen genomic analyses. Work is ongoing to overcome this, and to design studies to obtain adequate quantities of Pf DNA for these analyses. Such pathogen genomic studies are beyond the remit of the main trial protocol. We are confident that the varATS Pf qPCR assay will give more accurate estimates of population based prevalence in a post-intervention (MATAMAL) setting, using logistically feasible methods for sample collection (dried blood spots).

3) As in the previous point, do the authors have evidence the genotyping approach for different target regions from mosquitoes will work using an Illumina-based approach will work? They propose sWGA but this is very sensitive to DNA integrity so I am not sure of the feasibility. Apologies if this was unclear, Pf DNA will be taken from human dried blood spot samples, not mosquito samples: we have clarified the wording (p9, line382) to make this clearer. The cited references (Nag et al and recently published Moss et al[4]) report success using this method on hundreds of dried blood spots and RDTs from Bissau and the Bijagos, including samples collected over five years ago. We are confident this method is robust and appropriate for our trial, whilst also demonstrating the potential of a technology which is available in many malaria-endemic settings.

Reviewer: 3 Dr. Amber Kunkel, Santé publique France

Comments to the Author:

This protocol describes a cluster randomized control trial that is currently underway in the Bijagos Archipelago, Guinea-Bissau to evaluate the effect of ivermectin when added to mass drug administration with dihydroartemisinin-piperaquine. The primary outcome is the prevalence of malaria in each arm after two years of seasonal MDA (one month after the final round). This study could play an important role in improving our understanding of ivermectin's potential role in malaria control. There is a good evidence base suggesting a theoretical impact of ivermectin on malaria transmission, it remains unclear how well this approach will perform in the real world. Furthermore, this study's

design has several strengths that will help it to address this outcome, including the cluster-randomized design, the blinding of participants and investigators, and a setting which reduces the risk of cross-over effects.

The protocol provides a thorough literature review to support the importance of this trial, details on the setting where it is being conducted, and an overview of the primary and secondary outcomes including how they will be measured and assessed. My comments primarily relate to improving the clarity of the protocol. Additional review by a statistician with experience in analysis of cluster randomized control trials could be helpful, especially given the debate surrounding the analysis of the RIMDAMAL trial.

I would like to request the authors please review the entire protocol checklist and ensure all information is correct and complete, including page numbers. In several instances, the page numbers indicated do not appear to contain the specified information. For example, item #11C indicates page 14 but p.14 contains only references, and p.17 is listed as giving information on protocol amendments (#25) and plans for trial audits (#23), but I cannot find this information.

We apologise for these formatting errors and omissions relating to the SPIRIT checklist and have corrected all page numbers listed on the checklist. Supplementary materials have also been added and are fully referenced in the updated SPIRIT checklist.

The timeline of the study is a bit unclear. For example, dates are sometimes referred to by calendar year (ex. 2022, 2023), sometimes by reference to the year of the study (ex. first year, second year). I think better consistency here would improve readability. Furthermore, a more detailed timeline figure including calendar dates, years of the study, and timing of primary and secondary outcome measurements would be helpful. This could be either combined with or in addition to Figure 4.

2021 and 2022 have been replaced with "Year 1" and "Year 2" throughout, and are defined at first use (p6, line 240). A new timeline has been generated as suggested and included in Figure 1 (p5, line 196).

More information would also be useful for several topics including:

- Motivation for the choice of primary outcome

The aim of MATAMAL is to determine whether MDA with IVM and DP is more effective at reducing malaria transmission than MDA with DP alone. To evaluate this, it is necessary to select the most appropriate tool for quantifying malaria transmission. For this, it is important to consider the feasibility and cost of data collection and analysis, the sensitivity and specificity of the diagnostic tool itself, the reproducibility of methods and results, and the comparability to existing and future trials, among other considerations. qPCR prevalence was selected as it can reliably be obtained from dried blood spots, a cheap, simple and robust tool with good participant acceptability. It is also highly sensitive, even when detecting sub-microscopic infections. Laboratory analysis, though requiring highly-trained staff and expensive machinery, can be conducted at large-scale using existing facilities to which we have access at MRC The Gambia; this also helps to maintain these skills in a developing world setting. Furthermore, this methodology has been employed in other, similar trials, to which we would want to compare our results. The MASSIV trial was recently completed in the Gambia, comparing MDA with ivermectin and DP against no intervention, using a similar cluster-randomised design and using qPCR prevalence as the primary outcome comparator between arms. MATAMAL's methods and outcomes were deliberately harmonised to facilitate comparison. Among alternative measures, rapid diagnostic testing, for instance, is less sensitive, and entomological outcomes require unfeasible sample sizes and cost. Finally, the primary outcome was defined as "after two years," as the preparatory modelling demonstrated amplification of effect compared to one year, and two years of intervention still presents a reasonable timeframe for the conduct of a clinical trial. This information has been added to the manuscript (p6, line 273).

- Randomization procedure – how restrictive were the criteria applied? (e.g. of the 100,000 randomizations initially performed, how many met the criteria?)

The restriction variables used in the randomisation procedure are described in the manuscript (p5, line 191). Approximately 10% of the 100,000 randomisations met the criteria for inclusion. This figure has been added to the manuscript (p5, line 193). An independent statistician completed the final selection from those randomisations, so as to maintain blinding of trial staff.

- Procedures used to ensure data quality and plans to address any missing data

Data quality is integral to the outcome of the trial and we have taken steps to ensure high-quality data collection at all levels. A paragraph to this effect has been included on p9, line 420:

“Data will be entered by trained MDA delivery teams under the supervision of highly-trained and experienced local supervisors. Senior trial management will regularly review data entry and offer retraining as needed. Sample ID numbers will be electronically captured or double-entered. A communication network will be established with remote villages to facilitate monitoring and correction of errors found on review.”

Data will be collected through a mixture of paper and electronic forms, and all data entry will be performed only by trained personnel. Summary statistics will be reported daily by field teams to the site office to facilitate the early identification of problems. Databases will be cleaned and reviewed before any analysis. Ideally, gaps in data will be identified early enough to permit further data collection in the field to fill those gaps. We anticipate small amounts of data loss through, for instance, transcription errors of sample codes, but we have employed several methods, describe above, to reduce this risk.

- Informed consent process including assessment of potential benefits and harms and how these were communicated to participants. Would you be able to include forms used for data collection and informed consent as supplementary material?

We apologise for their omission in the previous submission; both consent forms (all ages) and data collection forms have now been included as supplementary material. The procedure for obtaining informed consent/assent is outlined in the manuscript (p5, line 220): following a structured community sensitisation programme, each household receives written and spoken information about the trial from their community health workers in their own language. Opportunity is given to ask questions and the trained workers return the following day to take consent. Participant information sheets list the potential harms and benefits, and these have also been included as supplementary material. It is made clear to participants that they can withdraw consent at any time without prejudice.

- Details on the data monitoring committee

The names of the members of the Data Safety Monitoring Board are now included as supplementary material. Their appointment was approved by our funders (in line with MRC guidelines) and independent ethical review committees. The overall aim of the committee is to safeguard the interests of trial participants, monitor the main outcome measures including safety and efficacy, and monitor the overall conduct of the trial. The committee will review data from the trial at least annually and make recommendations to the principal investigator. The committee charter can be made available on request.

- Are any analyses planned to assess the duration of the effect?

There will be no pharmacological analysis of duration of effect, but we will conduct an additional cross-sectional survey 12 months after the primary end-point survey (i.e. November 2023). This is additional to the trial (in scope and budget), and is therefore not included as part of the main protocol, but is clearly of interest to the MATAMAL investigators and the National Malaria Control Programme, with whom we work closely. This will provide valuable data on the long-term effect of the intervention and any rebound following cessation of an intensive intervention with respect to malaria epidemiology and vector ecology.

- What is planned for malaria control in this region when the trial is over – will any intervention be maintained? How will this be decided?

We work closely with the National Malaria Control Programme in Guinea Bissau, several of whom are co-investigators and partners for MATAMAL. With the Programme, we also work with implementing partners, such as the Global Fund, to understand the impact and implications of conducting this trial in the context of the Bissau-Guinean National Malaria Control Programme. Following the trial, results will be disseminated and shared with the Programme and relevant stakeholders and partners, including the WHO, in the context of current WHO recommendations to decide on the most appropriate control strategies beyond MATAMAL. We will continue malaria surveillance in the region, and hope to propose additional trials towards malaria elimination, potentially including promising interventions from MATAMAL depending on trial results. The region will continue using National Malaria Control Programme interventions, as it has done throughout the trial (p6, line 236) as a default.

- Please ensure abbreviations are defined before first use.

Apologies if some abbreviations were not defined at first use. We have revised the manuscript to ensure that this has been corrected.

We hope that our responses and modifications made to our manuscript satisfy the questions and concerns of the reviewers. We await their further review of our improved manuscript. Please do not hesitate to contact us if further clarification is required.

Yours sincerely,

Harry Hutchins
Lead author

REFERENCES

1. Slater HC, Walker PGT, Bousema T, Okell LC, Ghani AC. The Potential Impact of Adding Ivermectin to a Mass Treatment Intervention to Reduce Malaria Transmission: A Modelling Study. *J Infect Dis* [Internet]. 2014 Dec 15 [cited 2019 Oct 3];210(12):1972–80. Available from: <http://www.ncbi.nlm.nih.gov/pubmed/24951826>
2. Slater HC, Foy BD, Kobylinski K, Chaccour C, Watson OJ, Hellewell J, et al. Ivermectin as a novel complementary malaria control tool to reduce incidence and prevalence: a modelling study. *Lancet Infect Dis*. 2020 Apr 1;20(4):498–508.
3. World Health Organization. Updated WHO recommendations for malaria chemoprevention and elimination [Internet]. 2022 [cited 2023 May 17]. Available from: <https://www.who.int/news/item/03-06-2022-updated-who-recommendations-for-malaria-chemoprevention-and-elimination>
4. Moss S, Mañko E, Vasileva H, Da Silva ET, Goncalves A, Osborne A, et al. Population dynamics and drug resistance mutations in *Plasmodium falciparum* on the Bijagós Archipelago, Guinea-Bissau. *Sci Rep* [Internet]. 2023 Dec 1 [cited 2023 May 11];13(1):6311. Available from: </pmc/articles/PMC10113324/>

VERSION 2 – REVIEW

REVIEWER	Carrasquilla, Manuela Max Planck Institute for Infection Biology, Malaria Parasite Biology
REVIEW RETURNED	12-Jun-2023
GENERAL COMMENTS	The authors have addressed all comments. Thanks

REVIEWER	Kunkel, Amber Santé publique France
REVIEW RETURNED	11-Jun-2023

GENERAL COMMENTS	Many thanks to the authors for their detailed responses. I have no further questions. There do still appear to be a few issues with the page numbers in the protocol checklist, but perhaps this is just an upload or formatting issue. I look forward to reading the study results.
--